# Longevity as a Responsibility: Constructing Healthy Aging by Enacting within Contexts over the Entire Lifespan

**DOI:** 10.3390/geriatrics9040093

**Published:** 2024-07-12

**Authors:** Francesca Morganti

**Affiliations:** 1Department of Human and Social Sciences, University of Bergamo, I-24129 Bergamo, Italy; francesca.morganti@unibg.it or chl@unibg.it; Tel.: +39-035-2052060; 2CHL—Centre for Healthy Longevity, University of Bergamo, I-24129 Bergamo, Italy

**Keywords:** longevity, healthy aging, empowerment, flow experience, ecological lifespan development, enactive cognition, affordance, motivation, emotional selectivity, quality of life

## Abstract

Studying aging now requires going beyond the bio-psycho-social model and incorporating a broader multidisciplinary view capable of capturing the ultimate complexity of being human that is expressed as individuals age. Current demographic trends and the lengthening of life expectancies allow the observation of long-lived individuals in full health. These super-agers are no longer an exception. Indeed, individuals can have a good quality of life even over age 70 and living with chronic or neurodegenerative diseases. This change is driven in part by the cohort effect observed in people who are about to age today (e.g., better schooling, more advanced health conditions, and technologization) but more so by the gradual overcoming of ageist views. An aged person is no longer seen as a quitter but rather as one empowered to direct their own trajectory of potentially healthy longevity. According to this vision, this article proposes a situated lifespan perspective for the study of aging that integrates pedagogical models of developmental ecology with psychological theories of optimal experience to understand the individual motivational perspective on aging. At the same time, it does not disregard analyzing the daily and cultural contexts in which everyone situates and that guide aging trajectories. Nor does it forget that aging people are body-mind (embodied) organisms that, with contexts and through motivations, seize opportunities for action (affordances) to evolve in an optimal way during their lifespan. This theoretical reflection sheds new light on the aging process and on future trends in healthy longevity research.

## 1. A New Look at Growing Old

To study aging is to accept the challenge of understanding the ultimate complexity of human beings [1]. Facing this challenge requires working on multiple dimensions and contemplating not only multidisciplinary but, above all, interdisciplinary skills. The bio-psycho-social model [2] with which the phenomenon of aging has so far been investigated can be considered partly outdated, or at least incomplete. To fully understand aging, one must begin to consider the new dimensions of the demographic trend and outline what some scholars call a New Map of Life [3], which is yet to be drawn. It is a map where the final part of life is no longer a limited period to live in while resigning one’s autonomies or setting aside one’s self-determination, and not exclusively a time when life necessarily becomes fragile. Therefore, what it means to grow old today is not an easy topic to address. 

What needs to be focused on are the different potential trajectories of aging, within which one can find frailties and pathologies (not always determined by organic malfunctioning, but also by impairments due to social isolation and/or economic disadvantage, demotivation whilst going through the last part of the life course, etc.) as well as trajectories in which aging maintains some capabilities of adulthood within different roles. Therefore, the analysis of the determinants of longevity increasingly requires a very specific research focus on these issues [4]. To date, research interests have focused almost exclusively on the portion of the older population with dementia or with chronic diseases that are not self-sufficient and that, while undeniably important, constitute about 10 percent of the conditions in which the aging population grapples with [5]. There is therefore a tendency to not fully consider aging in the absence of disease and with good prospects for healthy longevity. In this regard, a much-debated topic in recent years is that of active aging, which currently no longer seems to be sufficient on its own to define the determinants of longevity [6]. From this perspective, in fact, one implicitly continues to think of an aged person whose role is to devote themselves to daily rest, who “reactivates” and does something exceptional to keep him or herself young. This perspective partially influences what is now best defined by healthy aging [7]. 

To better understand these different possible trajectories, however, it is necessary to emphasize the need to think of aging as the lifespan accumulation of experiences. Aging also has to be considered the growth of context-based skills and the progressive complexification of the person. Today, in fact, it is precisely these trajectories of aging that are most emerging. Above all, aging needs to be placed at the center of research as a situated phenomenon. Aging is not an abstract and circumscribed process but a phenomenon closely linked to the change of the individual and the contexts in which they are grounded. Aging takes shape in a context capable of influencing it and vice versa.

Many models have helped define this view. Consider, for example, the Eriksonian theory that contemplates aging as one of the fundamental steps in restructuring the self [8]. Theorized by Birren [9], the idea of “pleiotropism” seeks to reconcile the biological-hereditary dimension of aging with the contextual dimension, according to which the genetic component assumes a fundamental role in the early part of the lifespan but is also capable of making completely different characteristics emerge in later life. Other perspectives on the development of the individual in adulthood and old age depart from the biomedical view and introduce a social dimension of aging. For example, for Elaine Cumming and William Henry [10], activities, social roles, and relationships lose importance for the older person, who withdraws from social life and abandons their expectations in preparation for the arrival of death. Finally, the more recent positions of authors such as Lemon and collaborators [11] highlight how aging is a period with variable trajectories, always aimed at maximizing the person’s well-being. In any case, Baltes [12] is owed recognition for introducing lifespan psychology: the consideration of aging as a continuum between past and future. The new ideas that began to take hold at the end of the last century, such as the concept of “reserve”—whether cognitive [13], motor [14], or defined through the term “crystallized abilities” [15]—give due prominence to the path of aging as a possible future stage in life. 

It is in this context that the present contribution begins. Thinking of aging as a situated construction of the person’s integrity through individual change and possibilities of interaction with the contexts that revolve around context, constitute, in turn, an indispensable element of it in order to age healthily. It is a pathway that can no longer be thought of as influenced by mainly “external” determinants (such as genetic inheritance and/or the contextual condition in which one finds oneself living). One must contemplate the active role of the person capable of orienting their lifespan trajectory by “seizing” opportunities from the surrounding environment; by adjusting one’s choices to the skills acquired in the life course; by having clear in front of them the responsibility to optimize individual and contextual possibilities to ensure healthy longevity. 

As the American Psychological Association [16] has pointed out in defining the competencies that well-being professionals will need to have whilst working with older people, the current and future generations of aging individuals bring with them radical changes related to having grown up in different contexts and cultural eras. Today, this demographic change makes the aged “misfocused” individuals, and this poorly specified conception of aging is partly useless in the exercise of identifying patterns of person evolution, contexts that could foster their quality growing old, and needs on which to build age-inclusive services and/or proposals. This is why there is a need to broaden the gaze and use multifocal lenses in studying the aging phenomenon. It is time to provide innovative approaches to constructing research and intervention perspectives that can maintain most of the over-65 population as protagonists in the coming decades. 

The concept of responsible aging becomes crucial, and the path to becoming aged requires supporting and guiding a continuous empowerment of the person [17] over the lifespan. This is a good starting point for understanding how, in aging, it is necessary to no longer have a passive role. Rather, it is compulsory to take the reins of an individual life trajectory that becomes increasingly broader. Accordingly, responsibility in healthy aging implies that aging is no longer seen as the imminent approach to an abyss that inexorably leads toward death through loss of autonomy and frailty [18]—an idea unfortunately difficult to avoid since the stereotypical view of aging is often culturally linked to “ageism” and focused on older people with a loss of capabilities and a chronicity of disease [19]. Ageism is also a view according to which people after age 65 are called upon to play an increasingly less central role in today’s highly performance-oriented society [20]. Moreover, ageism today seems to fit tightly with most of the aging population [21] and undermines at its base the exclusively caregiving approach through which we address only the frail aged person or those with pathology, while forgetting those who aspire to healthy longevity. Accordingly, this stereotypical cultural dimension of aging must be overcome [22]. For healthy longevity, it is also necessary to create a society in which people of all ages can play an active role in the community. This implies that all people are called to individual empowerment about the role they can play in the community regardless of their age. Not many years ago, the dominant culture envisioned that older adults needed only an environment in which they could relax after the exertions of their working years and/or be cared for by their family members, if ill or in a frail condition. Today, the same community must come to terms with a new scenario and with increasing numbers of people living alone or who do not conform to this idea of rest, rejecting the need for dependence on younger generations. Addressing this necessary cultural change is not, however, so immediate. On the one hand, it requires that the possibilities for defining the potential of individuals must be reconsidered, and that the enhancement of instances and aptitudes must be initiated regardless of age. On the other, it implies that the full involvement of the community in the improvement of living conditions, and in the creation of a local community in which people at any stage of life can concretely envision their fulfillment, should be envisaged. By adopting this perspective, it becomes necessary to change the shared view that aging necessarily equates with loss and to open up new spaces of discussion in which to include development possibilities for aging people as well—possibilities providing them with opportunities for affirmation, learning, and sociality. It means, above all, recalibrating the significance of aging by guiding people to understand that growing old can mean having a fulfilling life in full quality.

There is an increasing urgency to address the need for a broader model that can enable a full understanding of what is happening to the new generations of older people and how best to support this umpteenth change in the lifespan. The main purpose of this article is to highlight how, by adopting an enactive and developmental ecology perspective, a vision of healthy aging can be better defined. The following paragraphs outline a multidisciplinary and interdisciplinary approach to healthy aging and some examples that arise, with the goal of approaching this definition.

## 2. Situated Aging: Elements That Can Contribute to Defining What Healthy Longevity Could Be

An attempt is made in this section to outline a multidimensional and multidisciplinary framework through which it becomes possible to study aging while respecting its complexity. The elements that contribute to constituting it are Urie Bronfenbrenner’s ecological theories of development [23], Mihaly Csikszentmihalyi’s flow theory [24], Paul and Margret Baltes’ Selection Optimization Compensation (SOC) model [25], Laura Carstensen and associates’ socioemotional theory [26], and the new models of aging in place observed by Camilla Lewis and Tine Buffel [27], the combination of constitute, in the present author’s view, the interpretive key needed to study aging and coming-of-age individuals. Individuals, as hypothesized by enactive theories of cognition [28], are body-minds in a continuous reciprocal relationship with affordances [29] that contexts offer as possibilities for a responsible evolution throughout the lifespan [1].

### 2.1. What Contexts? An Ecological Approach to Aging

First, it is necessary to remember that aging individuals are evolving organisms. They should be viewed in this way—not as static photographs of what one has become as an older person but as a result of change that has seen them undergo continuous remodeling in their own ecosystem of development. The specialization of an adult nervous system is an example of this. It is not the result of its linear evolution from an observable neural structure in childhood but rather a developmental process that takes place over time and in the places of human action [30]—a specific lifespan developmental trajectory that will originate a highly plastic system. It starts from a genetic predisposition in childhood but which in adulthood and old age will bring forth cognitive abilities and behavioral possibilities that are highly peculiar for everyone and influenced by the stimulations caught by the environment throughout the life course. The specialization of an adult individual is not the result of their linear evolution from childhood but rather a developmental process that is complexified in a nonlinear manner and that takes place over time and in places of human action. It is a specific developmental trajectory that will originate a highly differentiated individual in old age, starting from an initial predisposition that will bring out, during the life course, cognitive skills and behavioral possibilities shaped by the environmental and contextual stimulations experienced by the individual. Therefore, individual–context interaction becomes not only necessary but essential in defining individual, social role, institutional constraint, and cultural characteristics and how these interactions contribute to the peculiar development of that individual (Figure 1). For this reason, analyzing the micro-, meso-, and macro- contextual systems that are “nested like Russian dolls” and surround the individual and accompany them in their evolution over the lifespan [23,31] must be a cornerstone of research on becoming old. 

Whenever an aged person becomes the object of study, one must consider the individual as the center of a complex relationship of contextual systems in which they are the protagonist with their own distinctive traits (e.g., personality, habits, and motivations). At the same time, one has to keep in mind that that same individual participates in contextual systems in different ways. The same old person, for example, could be a grandparent who accompanies their grandchildren to school, could equally be a retired teacher who continues their volunteer work while still assuming an institutional role such as that of an educational tutor, could be a patient who goes to their doctor for an evaluation, or finally, could be a person who spends their free time with friends. Each time, this individual is within different micro-contextual systems and expresses different parts of their self. Therefore, to understand and assess individual potentialities and needs, one cannot forget to analyze which microsystem, at that moment, is engaging the individual. It is from the interplay between the individual and the different microsystems (which we can call a mesosystem), in fact, that one would be able to understand the possibilities facing the individual as well as the trajectories of their possible evolution in growing old. An example immediately emerges: when we consider the aging in place perspective, the exploration of the city takes on different forms (and the urban barriers became different) for the same individual when they are playing the role of aged patient on their way to the doctor, of aged friend who has to go to a meeting, of aged grandparent who is in the role of having to accompany their grandchildren on the home–school route, and so on. At the same time, it is likewise not possible to forget that individuals are surrounded by what is called the macrosystem, that is, the reference culture that frames every development and which, from time to time, determines how the individual is able to carry out the link with the context and the role they are assuming in it. There is no doubt that growing old in Italy is different from doing so in the United States and that having a surrounding culture that values the role of the old person (rather than an ageist culture that limits their possibilities) can lead to different aging trajectories in the same individual. 

Therefore, it is only by analyzing the aging individual in the different microsystems, and by reading the data collected in relation to the meso- and macrosystems, that one will be able to focus on what is the ‘growing up’ of that individual in the lifespan perspective—a perspective within which the person not only evolves but settles and stabilizes certain modes of functioning that then may become more or less useful to one’s healthy aging. It is from this point of view that what has been termed “reserve” (see [32]) must be studied and interpreted so that it can truly be an element in assessing that individual’s chances of healthy aging. Aging is not just as an accumulation of “neutral” experiences but as an expertise that has been built up clamped with the systems that that person has attended in their life course. Building up a reserve over the lifespan thus means not only having a “treasure trove of skills and knowledge” to use when resources eventually come to lack. It also means having planned to age by treasuring the possibilities and capabilities that are, from time to time, suited to the contexts in which one has lived. Today, there is a large debate about purely cognitive reserve as dysfunctional aging risk protection; motor and relational reserves are also essential. These reserves, together in their being exiguous when one grows older, determine potential vulnerabilities (e.g., risk of cognitive impairment, social isolation, motor frailty). It is for this reason that this article defines longevity as a responsibility to be assumed over the long term: the building of differentiated reserves are synergistic with the contexts they nurture through diverse experiences within the systems that individuals inhabit throughout their lives. It is these reserves, along with the motivations and strategies investigated below, that lead toward healthy longevity. 

In light of the foregoing, it becomes urgent to investigate how to activate the engine of a path of responsibility, which throughout life should have as its goal that of healthy aging through good lifestyles and the maximization of individual reserves, and how to maintain this propulsion without losing motivation in aging. This is especially so in cultures that tend to delineate aging as an inexorable loss and therefore demotivate accountability toward aging. It is, above all else, this long-term motivation that is lacking today in all those who fail to achieve what we call a “super aging” trajectory but that turns out to be very evident in those who brilliantly pass the 90–95-year mark while maintaining maximum autonomy and self-determination [33]. Thus, the questions to ask are as follows: What drives people toward this kind of evolution? If genetics matter only in part in longevity [34] and protective risk factors against chronic degenerative diseases are better understood [35], what motivations lead people to build healthy aging trajectories through lifestyles that lead them toward longevity in full responsibility? If individuals are evolving within developmental trajectories that they themselves choose (or avoid) and that will lead them toward different aging profiles, given equal resources, what motivates a person towards embarking on one developmental trajectory and not another? 

In an attempt to address these issues, research on reserve cannot be confined to the cognitive dimension but rather tether together the motor and relational dimensions within what is called a new form of active aging [36]. Likewise, the empowering perspective on active aging itself cannot be detached from analyzing the contexts in which it takes place, merging with the perspective of aging in place as starting to move away from a purely urban-centered view, yet beginning to consider places as opportunities for healthy aging [37].

### 2.2. What Motivations? The Responsibility for Healthy Aging

From the outset, individuals are guided in the development of their skills by a continuous interplay with the context they are immersed in and with those who co-participate in that context [38]. This means that everyone learns to evolve by selecting within contexts opportunities for action and by choosing what are purposeful behaviors in the contexts. Such behaviors will enable the individual to carry out actions by achieving a sense of perceived efficacy (agency) and to perceive the modification of the context itself as attributable to the consequence of their behavior [39]. It is in this way that, from childhood, every human being enriches their knowledge acquisition and that, in older adulthood, accumulates what is termed a reserve for aging. Motivation can be seen as the main driver of the continuation of this kind of skill acquisition path.

To understand motivation, one must consider what flow theory [24] states: People are able of being directed toward experiences of continuous agency and efficacy. It is the combination of behaviors and effects in contexts that guide the same individual in pursuing/eschewing experiences that return to the individual a feeling of effectiveness/ineffectiveness. Unfortunately, flow theory, which has been very successful in other areas of psychology, seems to be under-considered for aging (perhaps because, ageistically speaking, aging itself is not considered an optimal experience?). Instead, this article seeks to emphasize how, analogous to any other developmental stage, even in old age, the search for an optimal experience (comparable to flow) can be the driving force behind actions and behaviors aimed at change and continued learning of new skills, as well as motivation to age healthily (Figure 2). 

As defined in the flow, the optimal individual experience is given by the interplay of appropriate skills that the person assesses they have in order to deal with challenges that the context offers. Among the characteristics of the optimal experience, one finds a significant positive affect, including high levels of intrinsic reward, enjoyment, and, often, increased feelings of meaning and purpose. Thus, an appropriate balance between the context challenge and the relevant individual skillset enables the person to achieve flow by obtaining a sense of control (agency) and efficacy over the situation they are experiencing. Furthermore, according to recent scientific perspectives [40], the achievement of a state of flow corresponds to the involvement of large brain networks (e.g., the cortico-striatal-thalamic loop), as well as neurotransmitter systems (e.g., dopaminergic, noradrenergic, and endocannabinoid). These circuits seem to be amenable to the reward network that recent studies have shown to be partly modified in old age but with nevertheless an activation of the ventro-medial prefrontal cortex (vmPFC) [41]. One might therefore hypothesize that it is such motivating circuits, driven by the continuous search for a flow experience, that provide the individual with a strong drive toward self-efficacious and self-rewarding actions. Accordingly, a person who ages healthily is one who continues to be motivated to bring actions at play with respect to the contexts in which they are living and who, in doing so, seeks to achieve (by as much as possible) what is definable as an optimal experience. They do this by nurturing such a virtuous mechanism as long as they are able to succeed in keeping environmental challenges and individual skills in balance, in a continuous circuit of rewards for the self that takes on the characteristics of self-determination. This hypothesis is in agreement with Bakker and van Woerkom’s [42] proposal: there is a feedback loop in which flow experiences lead to better performance, which in turn feeds back into higher levels of self-efficacy and pushes individuals toward autonomy and responsibility. 

It is in pursuit of this approach that research to understand motivations for aging has been fielded in recent years [43]. By analyzing individual characteristics (demographic, cognitive, and emotional) and contexts (physical and cultural) in which people find themselves aging, and above all the combinations of individual and contextual characteristics, one can take a new look at healthy longevity looking at the situations of flow experience and hence at the determination for the healthy aging of both current aged and future aged populations.

### 2.3. What Strategy? Getting to Know Each Other and Evaluating the Challenge of Healthy Aging

The aging trajectory leads people to observe their abilities changing over time. Consequently, individuals must revise the challenge/skills balances that could determine an optimal experience. In aging, in fact, some of the elements of the relationship between contextual challenge and individual skills that determines flow must find new arrangements. In particular, the individual modifies their range of possibilities: Contexts evolve over time to require increasingly diverse skills. Prime examples of this are the introduction of new technologies and of the multilingualism through which knowledge is now spread. Both increase the difficulty for older people to participate effectively in daily life [44,45]. 

Regarding individual skills, substantial changes are observable in aging. Even if one does not go through decay or pathology, the aging body brings with it the gradual loss of some sensory, motor, and cognitive abilities. Furthermore, there are changes in socioemotional and relational styles (discussed in the following section). From a sensory point of view, reductions in visual and auditory acuity as well as changes in taste and smell can affect, for example, enjoyment of the surrounding space and nutrition habits [46]. Sensory systems can be considered the gateway through which the environment is perceived, understood, and evaluated by the nervous system, and they play an important role in shaping the cognitive abilities of people of all ages. Although effective corrective aids for some sensory systems (e.g., eyeglasses and hearing aids) exist, these are perceived as a sign of aging, and people often accept with shyness their inclusion in daily use. It is easily realizable how these modifications require reviewing the balance of challenges/skills that lead away from an optimal aging experience. In fact, the modification of receptive systems, the reduction in the speed at which they operate, and the quality of integration with other sensory, cognitive, and motor systems determine (from time to time) different responses to the stimulation of the environment, thus significantly impacting the maintenance of a good level of self-efficacy and autonomy. 

The cognitive capacity of the aging person brings with it transformations mainly evidenced in working memory, interference management, selective attention, and shifting the focus of attention from a narrower to a broader focus and vice versa [47]. These changes, mainly attributable to the despecialization of the central nervous system that follows an age-related posterior–anterior shift criterion [48], have consequences for daily coping skills. They may lead the individual to deal differently with new and old contextual challenges and require the substantial modification in their habits. For example, facing even a trivial everyday challenge, such as going out into a new neighborhood or following the operating instructions of a new home device, may involve finding oneself outside of that zone of optimal experience in which one has always found oneself. Being distant from such a flow, as described earlier, might cause the person to give up such behavior because they feel distant from a condition of effective agency. It is not uncommon that some changes in coping with everyday life stem precisely from this distancing from the condition of optimal experience. Accordingly, changes that lead individuals away from the feeling of efficacy in perceived agency drive those about to grow old toward a partial, but progressive and constant, renunciation of putting their abilities into play. This both demotivates the pursuit of empowerment to take an active part in building autonomy in old age and leads to the end of skill training and of acquiring new experiential knowledge useful in nurturing one’s reserve—a reserve that must be built in interaction with contexts and that, from time to time, allows one to travel along so-called successful and healthy aging trajectories. 

Thus, a variation in abilities is observed in aging (not change that necessarily goes toward a decline but of which aged people are aware to be evident in the aging of body and mind) that places individuals in a peculiar way to face contextual challenges. Accordingly, a polarization of aging is observable. From one side, the acquired awareness of how to cope effectively with aging changes, balancing in a new way the individual skills and contextual challenges by maintaining a daily state of flow that motivates the individual to take responsibility for their life path. From the other side, there is a newly acquired form of imbalance in the relationship between skills and challenges that increasingly leads the aging individual toward other conditions of experience referable to as “out of flow”. These out-of-flow experiences can include the anxiety to avoid aging at all costs (examples of this include some of the “anti-aging” trends found by researchers; see [49]) or the passive attitude of people who, facing even a temporary inability to cope with daily challenges, resort to the abandonment of responsibility (in which one glimpses so-called “introjected ageism”). Another frequent out-of-flow experience creates behaviors such as demotivation in following trajectories of potential healthy aging, like non-adherence to protective lifestyles, such as diet or restraint in the use of alcohol and smoking, which are now well known to prevent neurodegenerative diseases [50]. 

For healthy longevity to be achieved, solutions to these forms of challenge/skill imbalance can be multiple. On one hand, people can be educated toward responsible aging from early adulthood by promoting pathways aimed at enhancing the individuality of continuing education and psycho-health promotion [51,52]. On the other hand is the construction of practical and conceptual challenges that are more focused on the modifications of the aging person, as outlined by movements that reiterate not caring for including older people, such as in the age-friendly cities [53] and aging-in-place projects [54] increasingly pursued around the world. In both perspectives, the aim is empowering the aging generation to take an active and conscious role in the processes of change. 

That way, aged individuals do not give up on challenges and do not become demotivated in dealing with the events in everyday life that remain essential for growing old. In one definition, they become responsible for their longevity.

### 2.4. What Balance? The Chances of the Long-Lived Healthy

Healthy longevity is about being able to find an optimal balance between changes in individual capacities and the challenges that contexts present—challenges that become increasingly varied within different life contexts and contexts in which optimal experiences are sought; optimal experiences that motivate toward the perpetuation of a trajectory of responsible growth; and growth that does not stop with growing older. 

But what happens the moment this balance fails to be restored? In analysis of the life choices and strategies of healthy centenarians, or of those who have been able to optimally manage the changes that old age brings, one can observe the enactment of a pattern that has been extensively described in the literature and that unravels through three concepts: selection, optimization, compensation—SOC [25,55]. For example, the individuals who during their lifespan embark on trajectories of aging considered healthy are also able to select contexts, and parts of them, appropriately for the realization of their goals. As mentioned, perceiving the environment is not a “neutral” capacity; an individual is able to perceive the surrounding context in relation to their own self and the opportunities for action it offers (affordances). It is the affordances that, appropriately selected, enable an individual to maintain their responsibility toward a healthy aging trajectory, avoiding those opportunities that are more likely to prove unsuccessful and demotivate an empowered aging (Figure 3). 

One may not always find adequate challenges to one’s abilities. Optimizing one’s possibilities, patterned on the experiential reserve accumulated over a lifetime, leads an individual toward both selecting appropriate challenges and modifying barely reachable ones, or bringing into play additional individual resources that may improve the possibility of facing a challenge. It is not uncommon that people called “super agers” are those who have a good experiential reserve they can draw on and who are strategically able to bring parts of this reserve into play to cope with changes. They might be called optimizers of their own resources as well as the effective selectors of contexts in which they can still feel effective in a newly acquired balance of skill/challenge changes. 

It might equally happen that these people, despite the good selection and skillful optimization of the possibilities for interaction in contexts, need so-called compensatory processes. These could mean finding themselves in situations where there is a need to include people, aids, or procedures that can compensate for an acquired individual deficiency. Without compensation, the challenge one is about to face would appear largely unbalanced in its skills/challenge ratio. Today, bioengineering and technological advances, for example, present a plethora of new options for healthy longevity. These range from innovative services for transportation and communication to everything revolving around sensorics and robotics for diagnostics and health management. These innovations can ensure that the aged continue living independently using aids and services adapted to their needs. Compensation can also come in the form of caregiving that goes alongside people as they age, following trajectories that can be defined as healthy even though they require assistance. Even in the latter situation, it is necessary to reiterate that, for healthy aging, the ownership of caregiving rests with the individual themselves (see, e.g., the personal centered care approach [56]) and that this form of compensation does not demote empowered aging [57]. Whether environmental, technological, or human relationship elements, these forms of compensation should be seen as substantive changes that lead aged people towards a new equilibrium for which they themselves remain protagonists and responsible. 

Before proceeding further, it needs repeating that the introduction of the SOC model cannot be considered eradicated from the contexts in which each individual operates (previously referred to as micro-contextual systems) and, likewise, cannot disregard contemplating what influence these may have on the choice of compensations needed in meeting a challenge. As everyone who is about to age will have different microsystems in which they engage in behaviors or makes choices, the compensations chosen will strictly depend on what the specific context looks like to them and on the role that, within this context, the individual is assuming. Accordingly, the processes of selection, optimization, and compensation can be defined as situated, that is, closely related to the micro-contexts within which the individual is operating, from time to time related to the goal of the action that the person is carrying out within that context, and, finally, influenced by what the cultural macrosystem of reference is. It is, therefore, a model of selection, optimization, and compensation that is not defined “a priori” by the person. Rather, it is as a model of responsible aging that expresses all its value for a healthy longevity if actualized from context to context within this enactive and multi-componential vision. Thus, framing the SOC are the micro-, meso-, and macrosystems that guide the individual toward responsible and situated choices of healthy aging. 

Therefore, the most important educational work to be built with people about to age is to accompany them toward an ever-increasing awareness of the changes that aging brings and of the characteristics the contextual systems in which they find themselves aging so that they learn as early as possible to set up a situated SOC for healthy aging. 

### 2.5. What Perspective? Social Choices and Emotional Determinants in Approaching the End of Life

In any case, in dealing with aging, one cannot disregard that the prospect of death comes closer and closer in terms of subjective perception as years are added to age [58]. One hypothesis is that approaching death might modulate individual choices in different contexts, going on to influence how people select, optimize, and compensate for their chances for healthy aging. Moreover, the motivation to achieve an optimal experience can be interpreted through the filter of perceived closeness to death. Thus, further investigation into how emotions play a role in healthy longevity is absolutely necessary. 

The literature on emotions in the life cycle observes a change in “preferences” toward negative and positive valence emotions with advancing age. The negativity bias hypothesis emphasizes how younger people focus attention primarily on negative events in their lives by remembering them better than positive ones. This bias may occur because negative valence events are particularly influential during adaptation to context at this stage of life (e.g., development and reproduction). In contrast, the positivity effect hypothesis shows that older people remember positive events better than negative ones because they are aware that they will not live much longer and therefore tend to give less consideration to negative emotions and rely more on positive events from their past [59]. The latter perspective in particular has been extensively explored within the socioemotional selectivity theory of aging [60], which delves into the idea of time as a determining factor in guiding motivations and behaviors throughout the lifespan. According to this theory, time is not always experienced in the same way: it can be perceived throughout the lifespan as limited or expanded according to age. Thus, the time perception determines different ways of selecting goals as well as different motivations for pursuing them. The advantage of such an interpretation of time falls in the priorities with which social interactions are modulated (which are, in every moment of human life, the essential element to its growth), up to the point that motivations to act become motivations for social acting. Such motivations lead to the reshaping of possibilities/needs for increasing knowledge (e.g., understanding a new culture) and regulating emotions (e.g., sharing one’s moods) across the lifespan. Accordingly, the large body of research that Laura Carstensen has produced over the years [61] shows that, as people age, they tend to maximize the affective potential in social relationships, rather than cognitive potential. This prioritization of emotion over knowledge appears to be driven precisely by people’s perception of limited or expanded time with respect to their end of life. Thus, a person closer to the chronological age at which humans generally approach death will likely have greater motivation to direct themselves toward emotionally satisfying and meaningful activities (e.g., established friendships, and usual favorite activities). 

It is necessary to take this bias into account when analyzing older people’s tendency toward social withdrawal. In turn, the search for flow experience, and selections, optimizations, and compensations, could be influenced by this socioemotional distortion. It is not uncommon to find older people who give up on expanding their network of acquaintances, especially at a time when this could be functional and largely necessary for the reconstruction of a sociality that has unfortunately been lost through episodes of bereavement. Nor is it uncommon to observe individuals who tend to further forego empowerment in defining what are the necessary new caregiving relationships, as they do not consider them safe relationships (such as those exclusively limited to the family sphere, which is not always available). Such self-induced social isolation often adds a risk factor to those already present for the aging person, as it may lead toward an acceleration of the motor, functional, and cognitive decline as they remain isolated rather than gamble on emotionally unsafe relationships. This makes individuals less and less able not only to communicate with others but also to regulate their own motivational states in the intersubjectivity with others, and once again depleting a reserve that is not only cognitive but also fully relational. Lastly, considering that the context that revolves around older people proposes fewer and fewer opportunities for socialization and to the reactivation of networks of relationships, all of this can foster the tendency toward social isolation that older people already exhibit. 

It is quite easy to see how the risk of obtaining aging that moves away from the health trajectories discussed above greatly increases with this socioemotional selectivity bias. Hence, the emotions of aging, and how these can influence in their positive or negative valence the relational sphere that accompanies people’s growth trajectories, cannot remain a background element in understanding the pathways that guide toward healthy longevity. Once again, two elements are underestimated in aging research today, but they are essential. One is the culture (referred to earlier as the macro contextual system), which in environments where stable relationships, such as family relationships, are given a maximum value to determine an important weight in socioemotional selection. The other is the taboo of death, which in hedonistic societies is neglected and still constitutes a “hidden” variable in research on aging, despite the fact that it assumes more than a decisive role in the choices individuals made. Both have to be largely investigated and considered for healthy longevity.

## 3. Conclusions and Future Research Directions

This contribution has outlined how, in studying aging, it is necessary to embrace an enactive view of human cognition in which human development must be considered not as a pathway of accretion or loss of skills but as a continuous circular interaction with opportunities for action, provided by the environment, that can support complex situated developmental pathways. Growing old follows a line of ontogenetic development and can be considered the result of continuous adapted behaviors that take shape from the opportunities for action provided to individuals by the contextual changes surrounding them. Congruently with what is defined by the SOC model of aging, healthy longevity is the ability to adapt innate skills in a way that is, from time to time, more functional to environmental demands, which determine an effective path to being in the world [62]. Such abilities can certainly be defined as innate. At the same time, individuals’ innate abilities become effective because they are adapted to the demands that the context provides to them at a specific point in their lifespans. Especially in aging, this possibility of coexistence between innate and acquired components in human development underlines that an innate behavioral component may not manifest unless it is not conducting the individual in the ideal conditions to interplay with the environment in which they are placed. The environment, culture, and biological evolution, therefore, appear to be subject to the same rules in aging. On the one hand, in healthy longevity, the environment will have to be able to support, limit, or simply direct the expression a potential behavior presents from birth in the individual. On the other, we must increasingly become aware that what is biologically determined does not always remain invariable in its expression within an environmental context. Rather, it allows the individual optimal evolution by opening up the possibility of adapting every time to the affordances that the situation in which the individual is placed provides them. 

Thus, how should aging be studied if it is necessary to consider the mutual and progressive adaptation of the developing individual within environmental situations that are inherently changeable? First, by considering the aging individual as not an empty element that the environment can shape through stimulations and by studying how the context around an individual exerts its influence and requires mutual adaptation with the individual inhabiting it. Moreover, by focusing on a situated view of aging that contemplates the clamping between the individual and the environment and not limiting research to the mutual influences available in the here and now. The potential interconnections with multiple environmental situations that influence lines of development must be examined. For example, it is no longer possible to ignore the fact that people also use the environment according to their own needs and motivations. This perspective will be studied in more detail in the coming future. This ecological system, at the center of which the individual moves, consists of distinct and interpolated elements that can no longer be ignored. Aging will therefore no longer be a summation of experiences and abilities; individuals, contexts, and aging become closely related and interdependent. 

This is why the research streams developing in recent years within the group I supervise have accepted the challenge of placing the study of aging within the complexity model described so far. Among these, one project is accompanying medium-sized Italian cities as they become ‘cities of longevity’ by involving the population in a large-scale, multidimensional screening (which contemplates analyses of the cognitive, motor, and social reserves that people build) in order to support healthy longevity paths, possibly redirect trajectories of potential frailty in the aged, or protect against the risk of the onset of chronic degenerative diseases. This project, in the lifespan perspective described above, is not aimed exclusively at the aged population. It involves people aged 55 years and over who are recruited in their everyday environments (e.g., libraries, neighborhoods community centers, shops, and public places) and work environments (e.g., through the collaboration of companies and trade associations that insert the initiative as part of their health policies). The monitoring is to be conducted with the aim of understanding the potential trajectories of growing old, and for this reason it is not a simple early screening. Rather, in addition to the collection of quantitative data by means of standardized assessment scales, it includes a free interview that will be analyzed using a qualitative method in order to identify the emotions and motivations people have from the perspective of growing old. The data analysis will provide useful elements to understand the potential evolution of the prospects of aging, as well as to identify motivations, strategies, and socioemotional selections enacted by individuals within their own life contexts, so as to be able to educate them on a full empowerment toward healthy aging—even in vulnerable or frail conditions. Alongside this line of research, one project is complementary to it and adds investigative elements that expand the frame of the complexity of aging. In this project, which can be considered in the vein of aging in place studies, not only the physical characteristics of urban environments but also the opportunities for action that they offer to the aging individual are observed to understand how they can become constraints or possibilities for healthy longevity. As mentioned, not intending to consider the context in which people evolve as neutral, the idea of affordance as interplay between the individual and the environment, and its role for active and healthy aging, is analyzed herein. Accordingly, an urban ethnography (in which those over 65 are the main players, as they use their living spaces and neighborhoods) is conducted, together with a screening of the motor-cognitive-social reserves and the motivations that drive the experiences of the aged in certain places. Moreover, precisely in order to contemplate the different micro-contexts each individual inhabits in their development, the data will be interpreted in light of the different roles that the same agent may assume in relation to the context explored (as exemplified above, if they go out for a doctor’s appointment, to accompany a grandchild, to take a walk, etc.). This perspective definitely complicates the experimental design, but at the same time, it allows taking into account, and tying together, the developmental ecology model with the SOC perspective, allowing the idea of space as an opportunity for action for healthy longevity to emerge.

As these brief descriptions suggest (see [35,37,43] for details), all the projects to be carried out for the study of longevity originate from an enactive and situated dimension of human development. First, they focus on the characteristics of contexts within which the individual is growing old, of which they have direct experience, and which goes to constitute its activities, roles, and relationships, which are all experienced in that well-defined context. Second, they analyze contexts by referring not only to the objective properties presented by the environment but also to the meaning that the context has for the person experiencing it and to the possible relationships among the different contexts the person in aging in. Studying for the same individuals how the interactions between different environmental situations in which they take part can shed new light on how they participate in contexts of aging in different ways, on the value systems that govern the society in which they operate and experience, and on the beliefs and ideals that underlie their culture of reference, which may be directly or indirectly perceived by them and/or constitute elements for interpreting individual behaviors. 

Individuals and the time they spend directing growth trajectories in contexts generate such complexity that they cannot be regarded as equivalent. Each requires consideration as bearing peculiar individualities and study in their complexity over a period, spanning not only old age but the entire lifespan. Accordingly, it becomes impossible avoid considering human aging as a process of continuous and circular modification of skills acquired over a lifetime in perfectly situated manners. Skills that come to emerge within continuously acted interactions in contexts are not only physical but also culturally and socially characterized. 

Aging trajectories are increasingly seen as nonlinear, difficult to predict, and impossible to describe as inexorably “in” or outside the “norm”. Accordingly, studying the aging will recruit the interconnections between different contextual systems and people, who weave the web of meshes of the individual’s biological and psychological peculiarities, the physical and cultural contexts in which they operate, and the social situations and values that confront them, which will enactively codetermine the trajectories yet to be studied. 

In light of the foregoing, studying aging by focusing on what is happening today in the aging population and in the population that will age in the next 20 years (and that will make up the majority of the world’s population) means doing research interpolating the themes of neuro-constructivist and lifespan psychology, health and social economy, urbanism and aging in place, and finally the narratives and cultural constraints, such as ageist stereotypes, that together influence the trajectories of aging. The neuro-constructivist view presented here and interpreted in the lifespan perspective allows not only understanding the delicate balance between individual and context but also analyzing this balance in trying to understand how it can influence aging trajectories. 

At the same time, the concepts of cognitive and motor reserve, crystallized intelligence, and protective factors against chronic neurodegenerative diseases, now widely shared in the scientific community as determinants of healthy longevity, cannot continue to be ignored. The variability in terms of the health of individuals also results from their reserves and does not disregard the economic availability that individuals, due to their possibilities and/or due to change in the socio-economic conditions of the area in which they live, happen to have. Analyzing these factors allows an observation of the micro- and macroscopic influences of lifestyle choices that can affect what is called health literacy as well as compliance with healthy lifestyles within individual (e.g., chronic disease prevention) or community (e.g., risk protection policies) choices. 

In addition, the theme of aging in place and age-friendly city planning can really be a building block in the reflection on the role of the community effort for longevity if read through the multiple lenses of activities and roles that the ecology of development proposes. A city, like any other environment, is a place of affordances that are captured depending on the systems to be analyzed: places for work and leisure that change depending on whether one is taking on the role of professional, retired, parent, grandparent, or grandchild; places that, as such, need to be observed and possibly rethought to ensure that everyone can grow old without necessarily having to change places. 

Finally, it is impossible to ignore how the shared view of aging has brought with it an interpretation of old age as loss and decline and how this has contributed to the neglect of many of the alternative dimensions of aging. The ageist conception that has acted in determining the policies of engagement (rather non-engagement) of older people in places of self-determination of ideas is still far from disappearing. But, becoming more and more aware of the influence of stereotypes, such as educating for the non-denial but empowered view of people who are about to grow old, must be a revolutionary element in research and intervention with the adult and aged population. Accordingly, it will be possible to leave the dimension of service and care dedicated to that (minimal) part of the frail and/or pathological aged population and start to think about the effective projects and programs to support and sustain those (the majority) who yearn for healthy longevity. A small revolution is needed in the study of aging, one that leaves behind simplistic or compartmentalized views of aging and embraces a complexity of human becoming that cannot disregard its enactive and situated determinants.

## Figures and Tables

**Figure 1 geriatrics-09-00093-f001:**
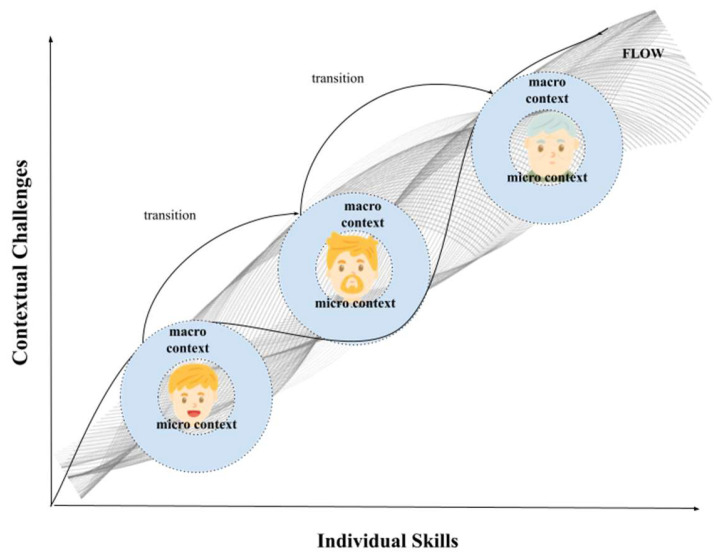
Illustrative diagram of individual–context interaction throughout life. At each stage of life, the challenges provided by the microsystems within which the individual implements their skills vary, contributing from time to time to the evolution of the individual in their growth and aging process.

**Figure 2 geriatrics-09-00093-f002:**
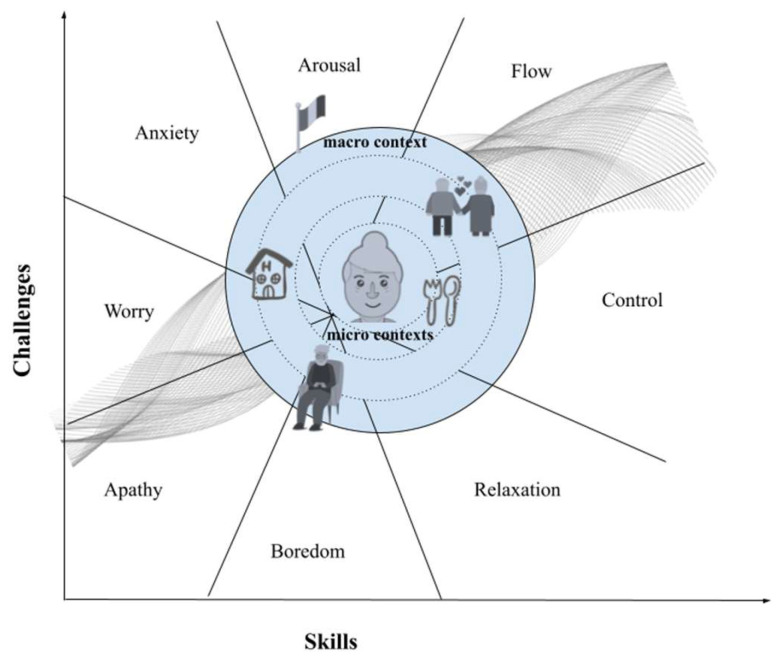
An exemplification of how the pursuit of a flow experience can be the motivational engine for embarking on a path to healthy longevity.

**Figure 3 geriatrics-09-00093-f003:**
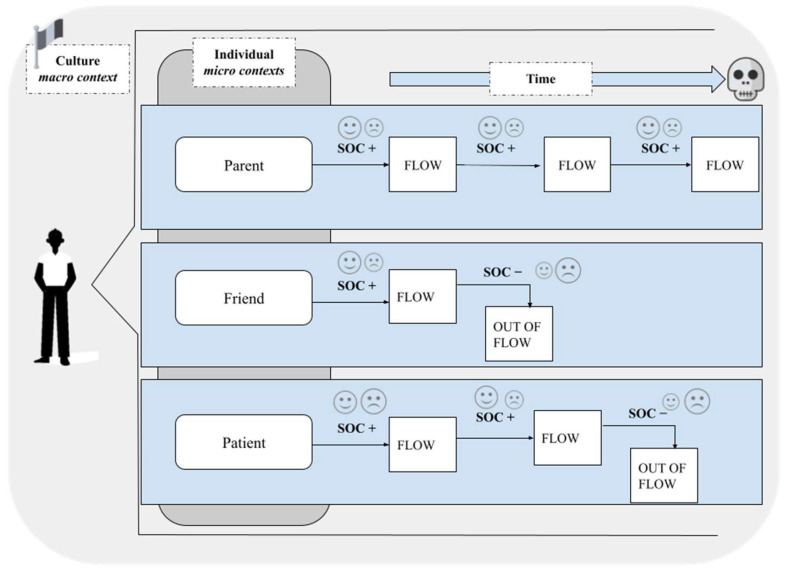
A flowchart in which the relationships among flow and out-of-flow motivations and SOC strategies are linked for healthy longevity pathways. The role of socioemotional selectivity according to death approaching is described in Section 2.5.

## Data Availability

No data were created or analyzed in this study.

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
