# Peer review of "Longevity as a Responsibility: Constructing Healthy Aging by Enacting within Contexts over the Entire Lifespan"

_geriatrics, 2024, doi:10.3390/geriatrics9040093_

Round 1

Reviewer 1 Report

Comments and Suggestions for Authors

The manuscript presents a comprehensive and multidisciplinary approach to understanding the phenomenon of aging, integrating pedagogical models of developmental ecology and psychological theories of optimal experience. It argues for a shift from the bio-psycho-social model to a more holistic view of aging as a complex and situated process, influenced by individual motivations, contexts, and opportunities for action.

Strengths

1) The integration of various theories and models provides a rich and holistic understanding of aging.

2) The concept of responsible aging and the emphasis on healthy longevity through active engagement and empowerment is novel and well-argued.

3) The emphasis on the role of micro-, meso-, and macro-contexts in shaping the aging experience is thorough and insightful.

4) The discussion on practical interventions, such as age-friendly cities and aging in place, offers valuable real-world applications.

Minor Revisions

1) Replace the term "elderly" with "older adults" throughout the manuscript as the former can be considered discriminatory.

2) Enhance clarity by simplifying complex sentences and ensuring key points are easily understandable.

3) Ensure that terms and concepts are used consistently throughout the manuscript to avoid confusion.

Detailed comments

1. A new look at growing old.

The introduction sets the stage well but could benefit from a clearer statement of the manuscript's primary aim.

Lines 61-63: Simplify the sentence for better readability.

Lines 117-125: when discussing about ageism, it is recommended to address the “Carta of Florence against ageism” (https://doi.org/10.1007/s41999-024-00938-7) and include it among the references.

2. Situated aging: Elements that can contribute to defining what healthy longevity 131

could be

2.1. What contexts? An 'ecological approach to aging.

The ecological approach is well-explained, but consider breaking down long paragraphs for easier reading.

2.4. What balance? The chances of the long-lived healthy

The discussion on SOC strategies is robust, but consider adding more empirical evidence to support the claims.

Line 403: [SOC, 24; 54] -> place the abbreviation outside from the square brackets that are used to identify citations.

3. Conclusions and Future Research Directions

Lines 582-590: The description of the project involving medium-sized Italian cities is interesting but would benefit from more detail about the methodology and expected outcomes.

The manuscript offers a valuable and innovative perspective on aging, emphasizing the importance of context and individual agency. With minor revisions, particularly regarding terminology and clarity, it will make a significant contribution to the field of gerontology.

The manuscript offers an invaluable and innovative approach to the study of ageing. Following the incorporation of a few minor revisions, particularly in regard to terminology and readability, the work will make a substantial and noteworthy contribution to the field of gerontology.

Comments on the Quality of English Language

Minor editing of English language is recommended

Author Response

Comment 1: Replace the term "elderly" with "older adults" throughout the manuscript as the former may be considered discriminatory.

I am sorry that I used the term "elderly". It was not my intention to use such a discriminatory word. On the 4 occasions I have done so, I have substituted "aged person" or "old". All substitutions are marked in red in the text 

Comment 2: Enhance clarity by simplifying complex sentences and ensuring key points are easily understandable.

Thank you for this comment. I have simplified the sentences make them more readable. You can fond the english editing in the _WITH CHANGES file. I hope the manuscript is now clearer to read

Comment 3: Ensure that terms and concepts are used consistently throughout the manuscript to avoid confusion.

The consistency of terms throughout the manuscript has been checked. changes soon indicated in red

Comments on " a new look in growing old":

1. The introduction sets the stage well but could benefit from a clearer statement of the manuscript's primary aim.

I have included in lines 127-129 a definition of the objectives (in red)

2. Lines 61-63: Simplify the sentence for better readability.

I added full stops e to simplify 

3. Lines 117-125: when discussing about ageism, it is recommended to address the “Carta of Florence against ageism” (https://doi.org/10.1007/s41999-024-00938-7) and include it among the references.

Thanks for the suggestion I added the reference in the text [22] and in the bibliography. All other references in the text have been updated in the sequence number between []

Comments on "Situated aging: Elements that can contribute to defining what healthy longevity 

could be"

1.  What contexts? An 'ecological approach to aging - The ecological approach is well-explained, but consider breaking down long paragraphs for easier reading.

I have divided the paragraphs for better reading 

2. What balance? The chances of the long-lived healthy

The discussion on SOC strategies is robust, but consider adding more empirical evidence to support the claims.

Line 403: [SOC, 24; 54] -> place the abbreviation outside from the square brackets that are used to identify citations.

Thank you for these comments. I added specifications for the SOC and removed the acronym from the parentheses

Comments on: "Conclusions and Future Research Directions"

1. Lines 582-590: The description of the project involving medium-sized Italian cities is interesting but would benefit from more detail about the methodology and expected outcomes.

Thank you mile for the very valuable comments that allowed me to improve the manuscript. I also thank the referee for appreciating the value of my contribution to the field of gentontology

Reviewer 2 Report

Comments and Suggestions for Authors

Dear Author,

your paper entitled "Longevity as a responsibility: constructing the healthy aging by 2 enacting within contexts in an entire lifespan." is a very interesting point of view investigating potential interplay between the  innate skills of a person and environmental challenges across the trajectory of ageing through lifespan.

Despite that I still  have some suggestions to make

1) environment is considered with a neutral charge and it is not subdivided in several aspects (societal, natural, financial etc). By doing this the author can add some useful insight in terms of suggesting interventions at the level of public health in order to make ageing process efficient for most people. By the current approach the whole process is mainly attributed to the persons capability of adapting its' skills to the changing environment which goes along with the personal responsibility for ones health issues. If you can add a short paragraph in this direction.

2) At the same time the text seems rather long and some sentences are longer than required and difficult to follow. Please simplify your text.

3) You use first person and this makes the text awkward. Try to change it

4) Some mistakes are there.

Line 103 ad maybe you mean and

Line 177 have to take in mind it is rather unlucky choice

Line 191 ethe maybe you mean the

Line 241 research on reserve cannot the correct is research on reserves

cannot

Line 326 327 the nervous system, and it play an important and it plays an important role

Line 343 have main consequences on daily maybe you mean many

You use aging and ageing in the text choose one

Comments on the Quality of English Language

Needs some improvement

Author Response

Comment 1 

Thank you for the suggestion, it was not my intention to define only the individual perspective to empowerment and certainly it is necessary to specify the dimensions of the context that contribute to this., I have added in blue ( lines 124-144) a short paragraph that I hope will satisfy the request 

Comment 2: At the same time the text seems rather long and some sentences are longer than required and difficult to follow. Please simplify your text.

I sent the manuscript to academic English proofreading to improve it you can find the new editing in _WITH CHANGES file

Comment 3: You use first person and this makes the text awkward. Try to change it

The text has been edited in the third person where necessary

Comment 4: Some mistakes are there

Thank you for spotting the typos, I have corrected them

Reviewer 3 Report

Comments and Suggestions for Authors

Responsible aging is a good concept. New map of life toward aging is a positive discourse. The purpose of this manuscript is:

“There is an increasing urgency to address the need to have a broader model that can really understand what is happening to the new generations of older people and how best to support this umpteenth change in the life span. For this reason, the following paragraphs will outline a multidisciplinary and interdisciplinary approach to healthy aging and some examples that arise with the goal of beginning to bridge this gap.” (lines 126-130)

There are many discussions in the text to focus on how to become the super agers. The multidisciplinary and interdisciplinary approach to healthy aging need to be pointed out more clearly.

Author Response

Thank you very much for your comments and for appreciating my contribution to the study of healthy longevity 

Here there is the revised manuscript including the referee 1 (in red) and 2 (in blue) comments